# 20 Years of ICF—International Classification of Functioning, Disability and Health: Uses and Applications around the World

**DOI:** 10.3390/ijerph191811321

**Published:** 2022-09-08

**Authors:** Matilde Leonardi, Haejung Lee, Nenad Kostanjsek, Arianna Fornari, Alberto Raggi, Andrea Martinuzzi, Manuel Yáñez, Ann-Helene Almborg, Magdalena Fresk, Yanina Besstrashnova, Alexander Shoshmin, Shamyr Sulyvan Castro, Eduardo Santana Cordeiro, Marie Cuenot, Christine Haas, Soraya Maart, Thomas Maribo, Janice Miller, Masahiko Mukaino, Stefanus Snyman, Ulrike Trinks, Heidi Anttila, Jaana Paltamaa, Patricia Saleeby, Lucilla Frattura, Ros Madden, Catherine Sykes, Coen H. van Gool, Jakub Hrkal, Miroslav Zvolský, Petra Sládková, Marie Vikdal, Guðrún Auður Harðardóttir, Josephine Foubert, Robert Jakob, Michaela Coenen, Olaf Kraus de Camargo

**Affiliations:** 1Neurology, Public Health, Disability Unit, Fondazione IRCCS Istituto Neurologico Carlo Besta, 20133 Milan, Italy; 2Department of Physical Therapy, College of Health and Welfare, Silla University, Busan 46958, Korea; 3Classification, Terminology and Standards Unit, World Health Organization (WHO), 1211 Geneva, Switzerland; 4Department of Conegliano-Pieve di Soligo, IRCCS E. Medea Scientific Institute, 31015 Conegliano, Italy; 5General Directorate of Health Information and Research, Ministry of Health, Mexico City 03100, Mexico; 6National Board of Health and Welfare, 10333 Stockholm, Sweden; 7Albrecht Federal Scientific Centre of Rehabilitation of the Disabled, 195067 St. Petersburg, Russia; 8Department of Physical Therapy, Universidade Federal do Ceará—UFC, Fortaleza 60020-181, Brazil; 9International Society of Experts and Researchers on Functioning and the ICF, University of São Paulo, São Paulo 05508-220, Brazil; 10School of Public Health, École des Hautes Études en Santé Publique (EHESP), 35043 Rennes, France; 11Jena University Hospital, 07743 Jena, Germany; 12Department of Health and Rehabilitation Sciences, Faculty of Health Sciences, University of Cape Town, Cape Town 7700, South Africa; 13Department of Public Health, Aarhus University, 8000 Aarhus, Denmark; 14DEFACTUM, Corporate Quality-Central Denmark Region, 8000 Aarhus, Denmark; 15Canadian Institute for Health Information (CIHI), Ottawa, ON K2A 4H6, Canada; 16Department of Rehabilitation Medicine, School of Medicine, Fujita Health University, Toyoake 470-1101, Aichi, Japan; 17Centre for Community Technologies, Nelson Mandela University, Gqeberha 6019, South Africa; 18WHO-FIC Collaborating Centre, South African Medical Research Council, Cape Town 8000, South Africa; 19The German Institute for Medical Documentation and Information (DIMDI), 51149 Cologne, Germany; 20Finnish Institute for Health and Welfare (THL), 00271 Helsinki, Finland; 21School of Health and Social Studies, JAMK University of Applied Sciences, 40200 Jyväskylä, Finland; 22Department of Sociology, Criminology, and Social Work, Bradley University, Peoria, IL 61625, USA; 23Classification Area, Azienda Sanitaria Universitaria Giuliano Isontina, 34128 Trieste, Italy; 24Faculty of Health Sciences, University of Sydney, Sydney, NSW 2006, Australia; 25National Institute for Public Health and the Environment, 3721 Bilthoven, The Netherlands; 26Institute of Health Information and Statistics of the Czech Republic, 128 01 Prague, Czech Republic; 27Centre Head of NordClass, Department of Classifications and Terminology in Healthcare, The Norwegian Directorate of e-Health, 0277 Oslo, Norway; 28Directorate of Health, 105 Reykjavik, Iceland; 29Census and Disability Analysis Office for National Statistics, Swyddfa Ystadegau Gwladol, Newport SA42, UK; 30Institute for Medical Information Processing, Biometry, and Epidemiology-IBE, Chair of Public Health and Health Services Research, LMU Munich, 80539 Munich, Germany; 31Pettenkofer School of Public Health, 80539 Munich, Germany; 32CanChild—Centre for Childhood Disability Research, McMaster University, Hamilton, ON L8S 4L8, Canada

**Keywords:** international classification, functioning, disability, health, ICF, public health, biopsychosocial

## Abstract

The International Classification of Functioning Disability and Health (ICF) was approved in 2001 and, since then, several studies reported the increased interest about its use in different sectors. A recent overview that summarizes its applications is lacking. This study aims to provide an updated overview about 20 years of ICF application through an international online questionnaire, developed by the byline authors, and sent to each World Health Organization Collaborating Centers of the Family of International Classifications (WHO-FIC CCs). Data was collected during October 2020 and December 2021 and descriptive content analyses were used to report main results. Results show how, in most of the respondent countries represented by WHO-FIC CCs, ICF was mainly used in clinical practice, policy development and social policy, and in education areas. Despite its applications in different sectors, ICF use is not mandatory in most countries but, where used, it provides a biopsychosocial framework for policy development in health, functioning and disability. The study provides information about the needs related to ICF applications, that can be useful to organize targeted intervention plans. Furthermore, this survey methodology can be re-proposed periodically to monitor the use of the ICF in the future.

## 1. Introduction

The International Classification of Functioning, Disability and Health (ICF) is the international standard for framing, describing, recording and measuring functioning and disability [1,2]. The World Health Organization (WHO) recommends using this classification alongside the International Classification of Diseases, 11th revision (ICD-11), that is used to report mortality and morbidity data [3]. ICF can also be used together with the third reference classification, the International Classification of Health Interventions (ICHI), to assess the needs for and to follow the results of performed interventions. Used together, these classifications can provide information about the health state of individuals and populations [4]. Due to the increased prevalence of chronic non-communicable diseases, ageing of population and a decreased mortality from infectious disease, there has been a change of perspective on health and disease scenarios worldwide [5]. This is the so called “epidemiological transition”, whose main implication is that data on morbidity and mortality alone are not sufficient to define the overall health status of populations, and therefore that an alternative concept of health is needed to represent it [6]. 

Health has been defined by the WHO since its constituency in 1948 but it is only since 2001 with ICF that WHO clearly defined disability within the frame of the biopsychosocial model. Disability is an umbrella term for impairments, activity limitations and participation restrictions, referring to the negative aspects of the interaction between an individual (with a health condition) and that individual’s contextual factors (environmental and personal factors) [7], which can act as barriers or facilitators [8]. 

In these first 20 years, two issues made clear the global importance and need of ICF. First, the need to have reliable epidemiological data on disability, which is essential at the global level as well as at country levels for public health issues and policy development. The WHO Report on disability states that over 1 billion of people in the word are living with some form of disability and this means that about 15% of the world’s population, with up to 190 million (3.8%) people aged 15 years and older have significant difficulties in functioning, often requiring health care services [9]. However, this number, which is expected to increase with epidemiological transition, is an estimate as only some countries are able to provide a precise number of who and how many people fall under this definition of person with disability, as definitions vary, thus lacking a common language. The second milestone since ICF adoption is the UN Convention on the Rights of Persons with Disabilities (UNCRPD), ratified in 2006, which defines, partially based on the ICF biopsychosocial model, that disability is not an attribute of a person, but rather a situation arising through interaction with different factors [7,10]. The cultural heritage of ICF in the UNCRPD is given by defining disability as “an evolving concept” but, at the same time, stressing that “disability results from the interaction between persons with impairments and attitudinal physical and social barriers that hinder their full and effective participation in society on an equal basis with others” [9]. The common conception of disability in the ICF and the UNCRPD makes the ICF the ideal tool for monitoring the influence and application of UNCRPD. 

The ICF is the result of global collaboration and numerous efforts made by academics, researchers, public health experts, clinicians and persons with disabilities providing a consistent and complete conceptualization of disability [2,7,11]. These 20 years have seen an increased use of ICF, as evidenced by numerous publications. Between 2001 and 2022, approximately 5600 scientific articles have been published with ICF (in full or as acronym) in the title or abstract and, in the last 10 years, 30 to 40 scientific publications have been produced each month on ICF. The first article about the possible use of ICF, in rehabilitation settings, was published by Üstun and colleagues in 2003 and reported that ICF is a valuable instrument for gathering and analyzing population health information from all around the world as well as a useful tool to collect data [12]. Two years later, a literature review exploring the main uses and the applications of ICF in different sectors showed that it had been applied for some clinical, social policy and statistical purposes, despite it being “just released” by the WHO [13]. Afterwards, two major reviews have highlighted the contribution of ICF in clinical practice and rehabilitation, as well as in non-clinical settings (e.g., education, employment, policy, technology development and statistics) [4,14]. The authors reported a global “cultural change”, due to the wide diffusion of ICF in research area with an important number of publications on scientific journal and its use in a great variety of fields [4,14]. Another literature review, conducted on health statistics area, reported the importance of ICF use for disability statistics and health information systems, defining the possible applications in national and international surveys as well as in national data systems for data collection [15]. Madden and Bundy, 15 years after ICF approval, in a scoping review, showed how ICF had contributed to creating a specific and standard language in the large panorama of disability, stimulating a change in thinking. However, the authors reported that the fields “need time to synthesize what has been learnt and further research and development is needed” [16]. The total amount of literature produced on ICF use and applications is very large and heterogeneous. However, despite this, the results address the potential rather than actual implementation of ICF. Hence, an update about the actual ICF use and implementation is needed to concretely understand where we are and which directions are we heading to [17]. 

This study was intended to address the state of the art of the ICF implementation 20 years after its adoption by reporting on the result of a global survey launched by the members of the WHOs Functioning and Disability Reference Group (FDRG). Specifically, the focus was on the main areas of use as identified by the members of the WHO Family of International Classifications (WHO-FIC) Collaborating Centers (CCs) network who participated to the survey.

## 2. Materials and Methods

### 2.1. The WHO-FIC CCs Network

The ICD, the ICF and the ICHI serve as the three reference classifications in what now forms the WHO-FIC, a network of CCs established in 1970 to support WHO’s work on international classifications. The principal role of the WHO-FIC Network is to promote the implementation, use, maintenance and updating of WHO reference classifications. The CCs in the WHO-FIC Network also assist WHO in the revision and development of the reference classifications. WHO Member States without a WHO-FIC CC can participate in the work of the WHO-FIC Network through technical representatives designated through their respective Ministries of Health. 

The WHO-FIC Network meets annually and progresses its work through Committees and Reference Groups, which conduct their business during and outside the annual meetings [8]. There are 27 CCs coming from all the WHO regions. Each WHO classification has a reference group with participants assigned by the CCs. The reference group for the ICF and related assessments is the FDRG [https://www.who.int/standards/classifications/who-fic-maintenance, accessed on 23 August 2022]. 

The FDRG launched an international online questionnaire to provide updated information on the use and implementation of ICF, as well as reported related issues and future suggestions. The questionnaire was sent to the WHO-FIC CCs and, although not all the countries are represented in the WHO-FIC CCs network, this approach can be replicated in the future with more countries. The web-based questionnaire was created and launched across 27 WHO-FIC CCs as the following WHO regions: Regions of the Americas (Argentina, Brazil, Canada, Mexico, United States of America) South-East Asia Region (India), Eastern Mediterranean Region (Kuwait), Western Pacific Region (Australia, China, Japan, South Korea, Thailand), European Region (Belgium, Czech Republic, Finland, France, Germany, Iceland, Italy, Norway, Russia, Spain, Sweden, The Netherlands, United Kingdom), and Africa (South Africa).

### 2.2. Survey Development and Administration

The survey was developed on the basis of previous literature and was grounded on the themes reported in the WHO’s and World Bank’s World Report on Disability [10] and on some of the most important pieces of literature on the ICF, including the first theoretical papers on ICF use [12,13,15], as well as on the previous literature reviews which identified with a bottom-up approach for the main areas of ICF use [4,14].

The survey was then finalized based on a discussion between FDRG chairs and members, and it was presented on occasion of the 2020 meeting of the WHO-FIC CCs, and the responses were collected from October 2020 to October 2021. Based on FDRG members’ discussion, it was decided not to pre-define any area of use (although relevant ones such as clinical use and use to inform policies were envisaged from the very beginning), but to rely on a bottom-up approach. The rationale for this was two-fold. First, to avoid replicating the results of previous pieces of literature and, second, to collect the perspective of the persons who have been working over two decades on implementing the ICF at different levels. This enabled highlighting the work that representatives made with policymakers, through clinical implementation, data reporting or more in general by promoting disability inclusion culture. Two sections were included in the survey (see Appendix B). 

The first section was directed to the countries where ICF was officially implemented and divided into 11 questions on: (a) the main uses of the ICF; (b) as a regulatory framework for documenting, coding and reporting functioning status; (c) level of coverage; (d) the data workflow for documenting, coding and reporting functioning data; (e) the quality of data coded with ICF; (f) important areas/data gaps/future data needs; (g) current human workforce and training requirements; (h) current Information technology (IT) infrastructures available to report functioning status data; (i) institution responsible for the maintenance of ICF in the country; (j) the main challenges on functioning status data in the country; and (k) about the impact and main challenges. 

The second section of the questionnaire was addressed to countries where ICF was not yet directly implemented (e.g., direct coding with ICF) and consisted of a single open-ended question in which it was indicated to specify the information requested in the previous section but using standardized or non-standardized data. 

### 2.3. Descriptive Content Analysis

For the purpose of the present paper, the focus was kept on the question on the main uses of ICF. Responses to the survey were managed by information saturation, for example each open-ended question was transformed into a closed option and frequencies were summed up by merging responses with the same content (e.g., clinical use cover response like “to address patient’s health status” or other like “to measure patients’ improvement over time”). Each time a new concept was found, a new option was added to the main question. A work of synthesis was thus made to avoid options that were similar: for example, clinical practice might cover sub-elements such as rehabilitation, pediatrics, neurology and so on. The intent was to report on the main areas: considering the relatively limited number of respondents (which correspond to WHO-FIC CCs), we could not have too much granularity (for more details see Appendix A). Once the single main questions were managed, we used frequencies and percentages to represent data. MS excel was used to manage data.

For those countries who did not report a formal ICF use and implementation, a brief description of the main areas was reported in a separate section (see Section 3.5).

## 3. Results

The survey was distributed to 27 WHO-FIC CCs. Responses were received from 20, at a response rate of 74% (Figure 1). Results showed that the ICF use was characterized by a strong heterogeneity, both in terms of the main areas of utilization and implementation levels, and in countries that were supported by a WHO-FIC CC as well. 

Only 14 out of the 20 respondent countries reported the current use of the ICF in at least one area with official support: by this, either a mandatory use or a strong commitment by users was intended. On the contrary, the remaining 6 countries declared that ICF use is still not mandatory, and few indications were proposed, either in terms of general knowledge of ICF, scattered experiences of use, or intentions for use where countries were heading to.

Figure 2 presents an overview of the main ICF uses among the 14 countries for which a clear and well-defined utilization was declared; Table 1 presents a detailed information on the main uses by country.

### 3.1. ICF in Clinical Settings

One of the main areas in which ICF was used by different countries was in clinical settings: where it was used for data registration in health care settings, outcome evaluation and for measuring disability accounted for 79% and 64% of the uses. Sweden and Australia reported the most widespread use in clinical settings (see Table 2). 

In clinical settings, the ICF was mainly used in a rehabilitation context and for outcome evaluation. The term “rehabilitation” comprehends not only the process to organize and plan a medical therapy or treatments, but also the evaluation of functioning status, the patient’s needs and the outcomes related. The majority of countries reported that the ICF was actually used as a reference model in the assessment of functioning, mostly in case of specific health conditions (e.g., after traumatic brain injuries, in disorders of consciousness, in stroke and in rehabilitation setting after hospital discharge). The ICF was also the reference model for Japan, South Korea and Canada, which used other tools to assess functioning, such as the Functional Independence Measure (FIM) [18] and the Barthel Index (BI) [19], which were not ICF-based. 

Other CCs reported the use of ICF in specific sectors of clinical settings: in social care for elderly and persons with disabilities, in registration of data of care and to assess workability. Some countries also reported its application in oral health, in the case of long-term illness or disability, in rare diseases and in municipal health care.

### 3.2. ICF in National and Regional Laws

Our survey showed that in all of the respondent countries, there were laws that imposed the use of ICF for classification purposes. Most countries did not have an official law that regulates the use of ICF as a classification and most reported that its use was not mandatory, despite a functional assessment being obligatory. However, there has been a global increase of implementation of ICF concepts, as this concept was reported by 57% of the countries, but actually the level of application remains low. Sweden, France, Italy and Australia reported the most widespread ICF implementation in national and regional laws (see Table 3).

Regarding the application of ICF in health and social policy legislations, the half of respondent countries (Germany, France, Sweden, Italy, Russia, South Africa, Canada) reported its implementation at regional and national levels. However, in most of these countries, the ICF was used as a general framework, and only in Germany it is embedded in legal health and social policies. In some countries, ICF is used for providing certificates assessing functioning (Sweden, Italy, The Netherlands, Australia, Russia), for health insurance coverage (France, Sweden, Australia) or for providing aids (The Netherlands). Despite the existence of recommendations in which individual assessment of functioning was mandatory and ICF was recognized as an official classification of functioning, its use was not.

### 3.3. Statistical Use of ICF 

Reports on the statistical use of ICF were available from CCs in Sweden, France, Italy, Russia, South Africa, Australia, Canada, Japan and South Korea (see Table 4). The highest level of implementation was reported for data collection through ICF-based tools (29% of the countries), with Italy, Australia and Russia reporting the most widespread ICF use. However, about 36% of respondents reported the use of instruments not based on ICF for collected data but which may be linked.

The data workflow varies by applications: some countries (Italy) reported the absence of data workflow, but the existence of a national repository of functioning data regulated by Ministry of Health (MoH). Other CCs (South Africa, Australia, South Korea) reported the use of ICF-based tools (e.g., ICan Function Mobile, ICare Government insurance and care schemes) for capturing ICF-related data. In other countries (Sweden, Finland, Russia, Japan, Canada), despite rehabilitation being provided within an ICF perspective, data are coded with another coding system. For example, in Finland, data on functioning in clinical practice and for research are collected by several measurement tools and they were linked to ICF by The Functioning Measure Database (TOIMIA) provided by the Finnish Institute for Health and Welfare. TOIMIA is an open access free-of-charge tool in Finnish, designed for experts and professionals interested in how to measure functioning in clinical practice and research, with the aim to unify the concepts used in measurement tools and harmonize assessment of functioning (ICF). It describes the psychometric properties, the recommended use and the linked ICF codes to the measurement items of over 120 functioning outcome instruments. Other countries (Italy, South Korea, Australia, South Africa) collect disability data based on some elements of ICF. In South Korea, for example, functioning and disability data were collected through the Korean Classification of Functioning, Disability and Health, which is a modified version of ICF and developed by Statistics Korea. However, it has not been implemented yet in any of the national statistical surveys.

In most countries (France, Denmark, the Netherlands, Czech Republic), functioning and disability data is not routinely coded; furthermore, no national data on quality of data was available. In other countries, the ICF, or parts of it, was used for data collection in national surveys (e.g., in France with its National Disability survey; in Australia with National Disability services data collection or Population Census). In other countries, it is used as a descriptor of disability at the regional level (e.g., in Italy through the Italian National Institute of Statistics—ISTAT) or to collect data at both national and regional levels (e.g., in Russia through the Federal Register of Persons with Disabilities).

### 3.4. Other Uses of ICF: Education (School System), Training on ICF and Research on ICF

Most countries (Germany, France, Sweden, Italy, The Netherlands, Finland, Australia, South Africa, South Korea, Japan, Russia, Czech Republic, Canada) reported the use of ICF in educational, training or research areas. In particular, Italy and Finland reported a full widespread use and research area was the one with the most frequent applications (see Table 5). These fields of application had already been reported in previous studies, and no new areas of applications beyond these emerged [4,14].

From the CCs responses to the survey, it emerged that the conceptual biopsychosocial model of ICF was widely used in education to understand the relationships between disease, impairments and activities and also their interaction with environmental factors. However, in most cases, the coding was not included in the education.

Italy and Finland reported that the individualized educational program for students with disabilities was modified based on the ICF version for Children and Youth (ICF-CY) and recently implemented in the education system. 

In South Korea, a web-based tool had been developed to assess functional status of students with disabilities for the educational support, whereas in Australia, ICF was proposed for use in designing resource distribution methods for school education.

Collaborating Centers participating in this survey report that the use of ICF was included in different educational programs at universities, with some in master’s degree curricula, training workshops or in dedicated conferences. Most of the CCs have developed one or more trainings on ICF and its use. The WHO-FIC CCs in South Africa hosts, on behalf of the WHO-FIC CCs and under the auspices of the FDRG and the Education and Implementation Committee, a web site on ICF training packages [20]. It is intended as a multilingual resource repository to be exploited by those developing ICF training materials, to exchange ideas and use available material; this enables increasing consistency in delivering ICF training. In 2001, Italy developed the ICF-basic and advanced courses that have been implemented by thousands of stakeholders in all different sectors. Many countries (e.g., France, Germany, The Netherlands, Finland and South Africa) reported that a training about the multidimensional approach of disability was mandatory for some jobs (e.g., rehabilitation medicine, occupational therapy or physical therapy). Italy also reported the use of ICF for training primary and secondary school teachers. 

Another use in educational areas was reported by CCs from The Netherlands and Canada in core curriculum design, for example, for nursing and allied health professionals (e.g., occupational therapy and physical therapy). A web-based training tool to teach the ICF model and its application, the ICF e-learning Tool, was developed by members of the German ICF Research Branch in collaboration with selected members of FDRG. Currently, the introductory module of the ICF learning Tool was available online in several languages (Danish, English, Finnish, French, Polish, Swedish), and other translations are in process. The ICF introductory module was useful for anyone interested in learning basic about the ICF and its application. The target audience of the ICF e-learning Tool includes the general public, people with disabilities, care providers and advocates, health professionals and people from the education and other sectors involved with services for people with disabilities [21]. A strong need for the education of health professionals and professionals from other sectors (e.g., education) on data collection and reporting information using the ICF and its coding system was evident from the survey’s responses.

Through our survey, different examples of ICF use in research areas emerged, with most countries reporting its application mainly in the fields of rehabilitation and education. International grants using the ICF, and its biopsychosocial perspective as a framework, have been reported by WHO-FIC CCs (EU projects MHADIE, MURINET, MARATONE, PARADISE, COURAGE in EUROPE) and several CCs reported that ICF was used in national and international grants as base for research development. Several studies were conducted in the WHO FIC-CCs for clinical reporting on specific health conditions, such as traumatic brain injuries, musculoskeletal injuries, hand injuries, almost all neurological and psychiatric disorders, and in many specialties. Applications was also carried out in specific settings, such as neuropalliative care, neurorehabilitation, speech pathology, physical rehabilitation, or child neurology. Some countries (e.g., Czech Republic, Finland, Norway, Russia, Australia, Brazil, Canada) reported the use of ICF-based tools, such as the WHO Disability Assessment Schedule (WHODAS 2.0), ICF Core Sets and ICF checklist to assess disability in clinical studies.

### 3.5. Initial Application in Countries Not Formally Using ICF

Some countries, specifically the United Kingdom, United States of America, Brazil, Mexico, Iceland and Norway, reported that ICF is not officially used, but nevertheless they reported the application of ICF or ICF-related instruments in different areas. Such experiences, which are still informal, thus represent the direction where these countries are heading.

In the UK, the ICF was not actively used within the National Health Service, however, the usability of ICF was examined in a number of projects conducted in clinical, educational, statistical and health and policy contexts. Those projects did not directly feed into official statistics, but helped developing insights into the role ICF can play in data collection, analytical research projects and policy interventions. Additionally, the USA reported no formal adoption of ICF, but some evidence indicates the use of ICF in education and practice were described. However, these uses were not communicated to the WHO-FIC CCs network, and an alternative mechanism for collecting functioning data was recommended and supported (surveying professional associations in the US). Brazil and Mexico reported the existence of specific laws based on ICF to guide social security planning, policy design and implementation. However, in both countries, ICF is not used as an official framework and information on data collection, data workflow and levels of implementation are lacking. In Iceland and Norway, the use of ICF was not mandatory and it was not yet implemented, despite having been translated and being available online on the official Directorate of Health websites, as well as used as a framework in national guidelines for rehabilitation.

## 4. Discussion

The aim of this study was to provide, 20 years after its approval, a global and updated picture of the main uses of the ICF, thus investigating within the WHO-FIC CC network if and how ICF has been implemented in various areas. The survey revealed significant variations in ICF use by the countries surveyed. These findings will serve as important baseline data to monitor the progress of ICF implementation over the coming years, as this can be considered a starting point to add information from countries without a CC but that are using ICF. This survey was a first step towards identifying the strengths and weaknesses of ICF and its implementation; this is of high relevance for the WHO in order to inform and guide the update and improvement of the classification, which is a process that never stopped and will not stop. Much of the scientific literature is on ICF applications in clinical contexts, i.e., production of lists of ICF categories and ways to implement it in different settings or with different types of patients. However, this embraces only a portion of ICF impact: changing the paradigm of how disability was perceived and addressed is the main achievement of ICF. Such a change moves from the theoretical perspective the ICF embraces, i.e., the biopsychosocial model, to finally get to the way in which the ICF can be implemented to make the paradigm change possible; this means implementing the ICF in administrative and statistical systems. However, this kind of information is hard to capture through an analysis of the published literature, and so our survey revealed a lack of capturing this information in administrative and statistical systems. The reasons are likely not technical, and since instruments to implement ICF exist, this is a further implementation step which needs to be targeted by countries and CCs.

In our survey, it was reported that the ICF has often been used as a conceptual framework and the main areas of application are health and social policies, clinical settings and education. Many countries reported an improvement in ICF use for coding functioning status but, 20 years after its approval, the implementation level remains low. In most countries, the use of ICF was embedded in health and social policies legislations, but despite this, ICF use was not mandatory in any of the respondent countries. The fields of application of ICF over the world were in line with its scopes and principles, but some important needs and issues emerged. In particular, the difficulties in the application of ICF codes and qualifiers, collecting data quality information or a lack of user-friendly electronic ICF-based tools (Appendix A). In addition to the issues that have been reported in our survey, it has to be acknowledged that over the last 20 years, several elements fostering ICF implementation have been produced. Examples of this include: a dedicated ICF checklist (i.e., selection of relevant ICF categories for specific conditions based on the analysis of sets of patients); ICF Core-Sets, which are now available for several conditions and settings of applications; ICF-based questionnaires and schedules; and different training modules. Therefore, if reasons for such a low level of implementation have to be searched, they are likely not technical but political. To overcome such a situation, a stronger effort of inclusion of a policy perspective into ICF implementation is needed: if we are able to provide clinical indications based on “evidence-based medicine”, we also need to learn how to prepare indication for decision makers so that they are able to produce “informed-based policies”.

One of the areas of application of ICF which has constantly been driving the process of ICF implementation is the clinical field. Those involved in ICF education have realized that the direct use of ICF categories and qualifiers would make ICF implementation complex and thus attempted to enhance the likelihood of ICF use by the development of tools such as ICF-derived checklists and Core Sets, ICF-based assessment instruments, and ICF-based assessment procedures. The ICF Checklist, a selection of ICF categories, is the most-used ICF-derived tool followed by ICF Core Sets, which constituted the first attempt to enhance ICF use and are developed with a standardized research procedure with the aim to guide data collection on functioning and disability [22]. ICF Core Sets are selections of ICF categories agreed on as relevant for specific conditions or clinical areas [23,24,25,26,27,28]. The most widely used ICF-based assessment tool is the WHODAS 2.0, used in a variety of setting and countries [29,30,31], but many others exist which are intended to measure disability in specific clinical populations: (e.g., patients with psychiatric and cognitive disorders [32], multiple sclerosis [33] or myasthenia gravis [34]). In our survey, some CCs report examples of ICF introduction in routine clinical activity, the most known being rehabilitation hospitals for either adults or children [35,36]. 

The use of ICF in the education sector can support continuity of information on functioning from entry at school and during the transitions from one educational level to the next or into subsequent work and employment. This continuity could be relevant for the pediatric population of the world as, different from the tools often used for functional assessment, such as Functional Independence Measure for Children (WeeFIM) [37,38,39] or Paediatric Evaluation of Disability Inventory (PEDI) [40], it provides information on impairments or limitations. The ICF can, moreover, combine disability information with other functioning elements important for learning (e.g., participation), thus improving the description of health conditions and impairment as well as identifying the key role of environmental factors. Furthermore, ICF provides the basis for goal-setting through supporting integration of assessment information from diverse sources, settings and perspectives. 

Training on ICF is also an important issue that emerged from the survey. Implementing global knowledge on ICF can contribute to strengthening health systems and the health status of individuals. Our survey suggests that continuous professional development programs for health workers and other relevant professionals should be improved. Online training such as the ICF e-learning Tool facilitates training activities independent from place and time and allows for saving personal resources.

In general, the information technology infrastructure available for documenting, coding and reporting functioning status remains poor and this was underlined by many of the respondents. The main challenges that emerge from our survey are: the low awareness at all levels about several benefits of ICF uses, low demand for data coded according to ICF, lack of economic motivation in reimbursement mechanisms for providers of health and social services and a strong need for national ICF-based rehabilitation standards. In addition to this, the fact that too few people are trained in ICF use and the fact that training materials and ICF-based tools (including their integration in existing information systems) are not user-friendly enough. These limitations, however, are part of the ICF implementation pathway and there are several elements that provide good reasons for continuing global ICF implementation. 

Most of the available literature on ICF is aimed to present ICF utilization in terms of ICF categories’ use to describe functioning and disability in a set of patients, either based upon a collection of clinical data or the definition of set of ICF categories through literature revision or expert consensus, i.e., the procedure of ICF Core Set definition. Conversely, to the best of our knowledge, few papers focusing on the main ICF uses have been published that address it in terms of areas. Exceptions to this include the papers from Ustun and colleagues [12], from Cerniauskaite and colleagues [4] and from Jelsma [14]. The authors of these manuscripts are from WHO or from the WHO-FIC CC network; this is likely the main reason for the correspondence on the main uses, in particular for the clinical and rehabilitation sectors, including the development of ICF-based instruments [4,12,14]. Our paper, on the contrary, presented the ICF-relevant areas of implementation together, whereas in previous studies these were jeopardized, despite being basically consistent: examples of this refer to ICF training [41], to the use of ICF to capture disability information [42], and to address disability eligibility [43]. 

Despite the ICF, as a classification, it has been approved by the World Health Assembly and is therefore an official document of WHO, and the amount of countries which actively participate to WHO-FIC CCs network is limited to 27 countries. These countries represent approximately 15% of the countries around the world, but more than half of the world’s population. However, out of these 27 countries, those who actually provided information account for only 14 of them of whom half are from Europe. Furthermore, those who responded to the survey are members of the WHO-FIC network: all of them are people who have been extensively involved in ICF implementation in their respective countries, and some of them in the early phases of ICF development as well.

This leads to considering the main limitation of this study; it was based on the availability of representatives of different CCs, and therefore, it should not be considered as exhaustive.

## 5. Conclusions

Although this survey is limited, as it concerns only the WHO-FIC CCs and the countries where they are located, it shows in fact that since its approval in 2001, ICF has been used not only in the CCs but worldwide as a global framework for describing functioning and disability. WHO may increase ICF use as it has now provided a new web-based version of ICF 2020, the latest and most complete version which overcomes the ICF-CY, now out of use, and includes categories relevant to the developing individual along their life span. The use of ICF 2020 allows countries to have an official version for which they can provide the translation. 

The use of ICF speaks to a real interest of those dealing with functioning and disability at the country level, and we hope that these use cases are a paradigm of a growing interest in the functioning of populations, as no other instrument is able to capture consistent and reliable data on functioning and disability as well as the ICF. Building on this survey, we hope that other countries will contribute to the collection of ICF use cases using the WHO collection tool, available on WHO website, to ensure its global relevance. 

Changing the paradigm of how disability is perceived and addressed is the main achievement of ICF; for a 20-year-old classification, this is a great success.

## Figures and Tables

**Figure 1 ijerph-19-11321-f001:**
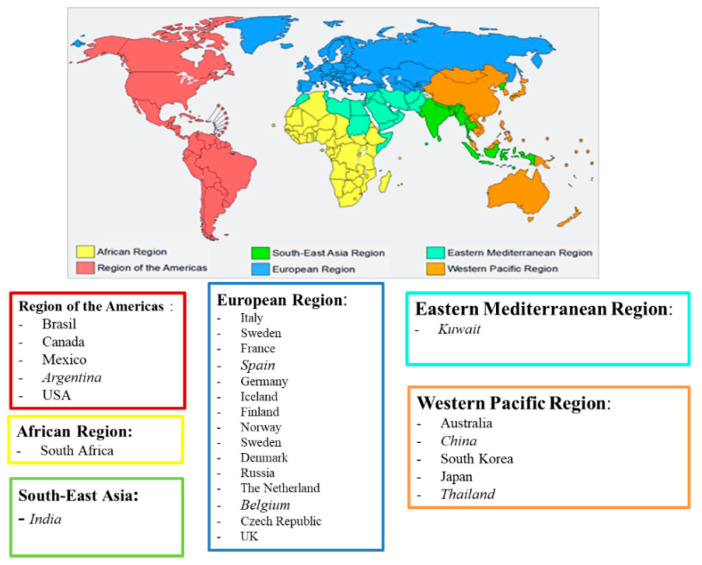
WHO Collaborating Centers that responded to the global survey. Legend: the countries reported in italics are those that did not respond to the survey (Argentina, India, Spain, Belgium, Kuwait, China, Thailand).

**Figure 2 ijerph-19-11321-f002:**
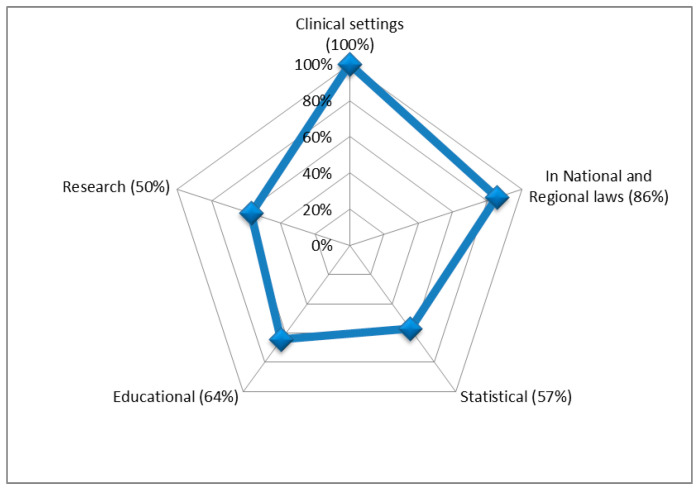
Main use of ICF in respondent countries (N = 14).

**Table 1 ijerph-19-11321-t001:** Main uses of ICF reported by WHO FIC CCs. Tables show the frequency of ICF implementation in different areas for each country.

Country	Clinical Settings	National and Regional Laws	Statistical Use	Educational	Research
Germany	√	√		√	√
France	√	√	√	√	
Denmark	√	√			
Sweden	√	√	√		
Italy	√	√	√	√	√
The Netherlands	√	√		√	
Finland	√		√	√	√
Australia	√	√	√	√	√
South Africa	√	√		√	√
South Korea	√	√		√	√
Japan	√	√	√		
Russia	√	√	√		√
Czech Republic	√				√
Canada	√	√	√	√	√

**Table 2 ijerph-19-11321-t002:** ICF uses in clinical settings.

Country	* Assessment Instruments	Rehabilitation and Outcome Evaluation	Oral Health	Rare Diseases	Registration of Data of Care	Municipality Health Care	** Social Care	Reference Model to Assess Functioning	Assessment of Workability	Summary Use by Country
Germany	√									11%
France		√	√	√						33%
Denmark	√	√						√		33%
Sweden	√	√	√		√	√	√		√	78%
Italy	√	√						√		33%
The Netherlands		√			√					22%
Finland	√	√					√	√		44%
Australia	√	√			√		√		√	56%
South Africa	√	√						√		33%
South Korea								√		11%
Japan		√								11%
Russia	√	√						√	√	44%
Czech Republic	√	√								22%
Canada								√		11%
Summary use by area	64%	79%	14%	7%	21%	7%	21%	50%	21%	–

Note. * ICF was used as needs or functioning assessment instrument. ** For elderly and persons with disabilities.

**Table 3 ijerph-19-11321-t003:** ICF uses in national and regional laws.

Country	ICF Implementation at Regional and National Level	ICF as General Framework in Social Policies and Legislations	ICF Use Embedded in Legal Health and Social Policies	Insurance Medicine Decision Support	Certificates for Assess Functioning	Providing Aids	Summary Use by Country
Germany	√		√				33%
France	√	√		√			50%
Denmark		√					17%
Sweden	√	√		√	√		67%
Italy	√	√			√		50%
The Netherlands					√	√	33%
Finland							–
Australia		√		√	√		50%
South Africa	√						17%
South Korea		√					17%
Japan		√					17%
Russia	√				√		33%
Czech Republic							–
Canada	√	√					33%
Summary use by area	50%	57%	7%	21%	36%	7%	–

**Table 4 ijerph-19-11321-t004:** ICF application in statistical areas.

Country	Data Collection (National and Regional Survey)	Data Collection (ICF-Based Tools)	Data Collection (Coding System not ICF-Based)	Summary Use by Country
Germany				–
France	√			33%
Denmark				–
Sweden			√	33%
Italy	√	√		66%
The Netherlands				–
Finland			√	33%
Australia	√	√		66%
South Africa		√		33%
South Korea		√		33%
Japan			√	33%
Russia	√		√	66%
Czech Republic				–
Canada			√	33%
Summary use by area	29%	29%	36%	–

**Table 5 ijerph-19-11321-t005:** ICF implementation in educational area, training on ICF and ICF use in research.

Country	School Systems	Training on ICF	Research on ICF in Specific Settings and Health Conditions	Summary Use by Country
Germany		√	√	66%
France		√		33%
Denmark				–
Sweden				–
Italy	√	√	√	100%
The Netherlands		√		33%
Finland	√	√	√	100%
Australia	√		√	66%
South Africa		√	√	66%
South Korea	√		√	66%
Japan				–
Russia			√	33%
Czech Republic			√	33%
Canada		√	√	66%
Summary use by area	29%	50%	64%	–

## Data Availability

The dataset supporting the conclusions of this article is included within the article and its additional file (Appendix A). All information, about database uses and analysis conducted in these study, are available from the corresponding author on reasonable request.

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
