# Peer review of "20 Years of ICF—International Classification of Functioning, Disability and Health: Uses and Applications around the World"

_ijerph, 2022, doi:10.3390/ijerph191811321_

Round 1

Reviewer 1 Report

The document aborts an important topic, however I have some observations:

It is not clear if it is an original investigation, bibliographic review or meta-analysis, since the application of a questionnaire is mentioned in the methodology and a section of a document search appears in the results.

The methodology is the section in which there is a greater deficiency and this is related to the fact that there is no clear objective of the research, for the analysis of the data, only a descriptive analysis is not enough.

Results, are not properly described with reference to the statistical analysis, there is no evidence of their analysis in a table or figure. This section should have 50% of the content of the document.

Ddiscussion, does not observe the comparison with other studies.

The conclusion is not punctual according to the objective, it is very extensive.

Reviewer 2 Report

The proposed paper ‘20 years of ICF - International Classification of Functioning, Disability and Health: uses and applications around the world’ reports the results of an investigative questionnaire, submitted to different countries across the world, about the nationwide use of the International Classification of Functioning Disability and Health (ICF) in different sectors, including legal frameworks, clinical practice, statistics, education and more.

The topic is of high relevance, presenting a follow-up about the actual implementation of a worldwide classification standard which held great potential benefits.

However, in my opinion, there are some issues to be addressed by authors before publication:

1) The main problem is language. The submitted text presents several errors, not only in terms of graphics (e.g. many references are outside brackets, especially in the discussion section) but also in the form. In particular, the use of commas is often incorrect, separating the subject of the sentence from the verbal predicate (some examples are in line 91, 105, 191). Many sentences should be rephrased (e.g. lines 94-98), as they are grammatically incorrect, and some typos are present even in the appendix, where the text of the questionnaire is reported (e.g. line 514). In the current form, it is uncomfortable to follow through the reading.

2) Data collection (from line 180): in my opinion, it would be important to include a brief description of the decisional process (who did it, how and why) for the definition of the questions to include in the questionnaire and, consequently, of the rationale that led the choice to focus on the aspects included and to exclude other possible perspectives.

3) Results section: a table or a schema should be added to report the main results at a glance. For example, one possibility is to present the number of countries reporting a specific use, subdivided by categories (law, clinical practice, statistics etc.), even though it implies a certain level of approximation compared to the cases described in the main text.

4) Discussion section: it would be interesting to present some insight about geographical differences, or continuity, across the different countries and the different areas. Moreover, authors should include some proposals (possibly practical, but also just theoretical) of priority action to help relieving the main pains identified.

5) Line 277 Figure 2: please add label numbers to the graph grid

Reviewer 3 Report

This manuscript describes the use of the International Classification of Functional Disability and Health (ICF) worldwide in the first 20 years after its introduction, using an online questionnaire. The study is of great significance in this area and the authors are to be congratulated for their efforts. Although some difficulties can arise in describing the results of an international survey in the context of different national backgrounds, I’d like to suggest the followings.

Introduction

This is valuable resource for the reader, with ample references and a good summary of the background to the ICF's formation.

-         L127 Please delete “,” after “use of ICF,,”

Abstract

1)     The abstract mentions descriptive statistics, but there seems less statistics in the text. If it is considered inappropriate to state frequency when the survey is covering CCs in each WHO region and some countries have not responded, the statement ‘descriptive statistics’ should be replaced to simply “descriptive analysis’.

2)     The Abstract states that "ICF is mainly used in health and social policy areas, while some regions reported its applications also in clinical practice and educational fields.", while in Figure 2, Clinical Practice is the highest. Readers may be confused by the way conclusions are drawn, so please describe clearly what has been known and how this survey could add the evidences.

Materials and Methods

1)     L171  It is assumed that the general reader is unaware of how WHO Collaborating Centres are selected and what they do. Therefore, a reference list or website should be appended for 27 Collaborating Centres among all WHO regions.

2)     L176, Parentheses for the literature are missing. research10,12-17 research [10, 12-17]

3)     L196 Descriptive analysis should be replaced to ‘Descriptive contents analysis’.

Results

1) The summary table should be created since the supplementary table has too much information. It is only indicated for the main uses for countries that currently use ICF in results part in manuscript. As mentioned above, there seems less statistics in the text. It would be informative if the authors could show the frequency on each item is implemented in the responding countries for section 1. Because readers would have difficulty grasping what the issues are from this survey. Therefore, the authors could categorize or classify ( yes/no/NA, less/moderate/well, etc) the following answers from b) to k).

(b) as a regulatory framework for documenting, coding and reporting functioning status,

(c) level of coverage,

(d) the data workflow for documenting, coding and reporting functioning data,

(e) the quality of data coded with ICF,

(f) important areas/data gaps/future data needs,

(g) current human workforce and training requirements,

(h) current Information technology (IT) infrastructures available to report functioning status data,

(i) institution responsible for the maintenance of ICF in the country,

(j) the main challenges on functioning status data in the country

(k) about the impact and main challenges

2) If there appears to be a separate official publication in a WHO report or elsewhere, please append it, and if it is not appropriate for publication in this paper (e.g. it does not fit into international comparisons), please explain why.

3)     L214-216 discusses Figure 2, but does not seem to adequately describe what the figure shows.

4)     L217. Main uses for counties that currently use ICF countries?

5)     L239. Please delete space before “for data”

6)     L243. Please remove ‘,’ before “for organizing a treatment plan,”.

7)     I would like to see it inserted immediately after mentioning the location of Figur1,2, but please discuss this with the editorial staff.

8)     The scale in Figure 2 needs to be filled in as it has not been filled in. The number of respondent countries (n=20) should also be entered. If there are duplicate responses, these should also be noted in the legends.

9)     The discussion is mixed with the results, which makes it difficult to distinguish the results from those derived from the survey. So please describe them clearly and separately.

ex. L367-368, L446

Discussion

1)     L390-391 “Many countries reported an improvement in ICF use for coding functioning status but, 20 years after its approval, the implementation level remains low.”

Please add comments and suggestion why the implementation level remains low

2)     Although mentioned in the text as a limitation, please discuss the representativeness of the data from the Collaboration Centre in this study.

3)     Do the attributes of the respondents have any influence?

4)     Parentheses for the literature are missing in Discussion. L406 27, 2833.etc

5)     Discussion is needed on countries without WHO CC and on South-East Asia and Eastern Mediterranean Region where no responses were received in this survey.

Reviewer 4 Report

This article presents research with the aim to provide overview of ICF application during the last 20 years. These my comments and suggestions:

Abstract:

Please correct errors like " increased interest about it use"....its use, I presume. There are some spaces in front of the comma in keywords part.

Introduction:

Please correct sentences and typographical errors, e.g.: "Used together these classifications, can provide information about the health state of individuals and populations [4].", " increase use of ICF,,", etc.

The aim of the research should be stated more clearly, please rephrase. Explain all abbreviations used (e.g. FIC, CC).

Materials and Methods:

Correct errors like this: "Based on previous research10,12-17". Methods should be more specific and contain more details following STROBE guidelines.

Results:

In the Methods section you stated that "we used frequencies
and percentages to represent data.". I do not see any percentages in the results section.

Discussion:

There are many typographical errors in this part of the manuscript, e.g. "(e.g. patients with psychiatric and cognitive disorders36,or multiple sclerosis38or myasthenia gravis39)". Please, correct them. Discussion should be stronger. Explain why is your research important and what is the scientific contribution of the study.

Conclusions:

Your Conclusion chapter is far too long.

Round 2

Reviewer 1 Report

Dear authors, the document has been improved, I only have a general observation, throughout the document there are very long paragraphs, authors should separate these paragraphs by ideas and with no more than 6 lines, it would be very tedious for readers to follow the central ideas.

Author Response

Dear Reviewer, thank you for your comment. We tried to revise the manuscript accordingly to the best of our ability and you can see changes in the manuscript in yellow. We also reported the text between squared brackets, showing each sections where we split sentences to reduce their length.

Best wishes, the authors

1) Dear authors, the document has been improved, I only have a general observation, throughout the document there are very long paragraphs, authors should separate these paragraphs by ideas and with no more than 6 lines, it would be very tedious for readers to follow the central ideas.

R1) We are glad you are satisfied with the revision: now we tried to reduce the longest sections. Having paragraphs of no more than six lines is of limited feasibility and therefore we focused on the longest sentences (some of whom were much above 5-6 lines). In the manuscript you can appreciate the sentences that have been split, the one highlighted constitute the “second part” of a previous longer sentence. Please see changes on lines:

  • 133-136 [Two years later, a literature review, exploring the main uses and the applications of ICF in different sectors showed that it had been applied for some clinical, social policy and statistical purposes, despite it was “just released” by the WHO [14].]
  • 201-203 [This enabled to highlight the work that representatives made with policymakers, through clinical implementation, data reporting, or more in general by promoting disability inclusion culture.]
  • 316-318 [In other countries (Sweden, Finland, Russia, Japan, Canada) despite rehabilitation is provided within an ICF perspective, data are coded with another coding system.]
  • 336-339 [In other countries, it is used as a descriptor of disability at regional level (e.g., in Italy through the Italian National Institute of Statistics – ISTAT) or to collect data at both national and regional levels (e.g. in Russia through the Federal Register of Persons with Disabilities).]
  • 346-348 [In particular, Italy and Finland reported a full widespread use and research area was the one with the most frequent applications (see table 5).]
  • 368-371 [It is intended as a multilingual resource repository to be exploited by those developing ICF training materials, to exchange ideas and use available material: this enables to increase consistency in delivering ICF training.]
  • 379-381 [A web-based training tool to teach the ICF model and its application, the ICF e-learning Tool, was developed by members of the German ICF Research Branch in collaboration with selected members of FDRG.]
  • 399-401 [Applications was also carried out in specific settings, such as neuropalliative care, neurorehabilitation, speech pathology, physical rehabilitation, or child neurology.]
  • 458-460 [In particular, the difficulties in the application of ICF codes and qualifiers, collecting data quality information, or a lack of user-friendly electronic ICF-based tools (Table S1 in supplementary materials).]
  • 462-466 [Examples of this include: dedicated ICF checklist (i.e. selection of relevant ICF categories for specific condition based on the analysis of sets of patients); ICF Core-Sets, which are now available for several conditions and settings of applications; ICF-based questionnaires and schedules; different training modules.]
  • 494-497 [The ICF can moreover combine disability information with other functioning elements important for learning (e.g. participation), thus improving the description of health conditions and impairment as well as identifying the key role of environmental factors.]
  • 511-514 [In addition to this, the fact that too few people are trained to the ICF use and the fact that training materials and the ICF-based tools (including their integration in existing information systems) are not enough user-friendly.]

Finally, kindly note that two long sections cannot be reduced: one is on lines 180-186 where we report the countries with collaborating centers (as per a previous comment) and another is on lines 205-213 where we report the elements of the survey.

Reviewer 4 Report

I am satisfied with the improvements of the manuscript. However, scientific contribution of this manuscript is not major.

Author Response

Dear Reviewer, thank you for your comment. We tried to revise the manuscript accordingly to the best of our ability and you can see changes in the manuscript in yellow. We also reported the text between squared brackets.

Best wishes, the authors

1) I am satisfied with the improvements of the manuscript. However, scientific contribution of this manuscript is not major.

R1) We are glad you are satisfied with the revision: it took a lot of effort, also in trying to better highlight the scientific worth of our manuscript. The point, which we try to resume, is that if you think back to the 20 years of ICF, what you see in the literature is a lot of manuscripts which present lists of ICF categories and ways to use in the clinical field. This however represents only a portion of ICF implementation and impact, which embraces the contribution to the theoretical field, i.e. the change in paradigm how is disability perceived through the biopsychosocial model, but also the way in which the ICF can be implemented to make the paradigm change possible: this means implementing the ICF in administrative and statistical systems. And this is the point we highlighted as the major weakness, also pointing out that the reason for ICF implementation is political rather than technical.

We tried to re-phrase the section at the beginning of discussion to make the scientific relevance of the paper more evident, see changes on lines 438-450 [Much of the scientific literature is on ICF applications in clinical contexts, i.e. production of lists of ICF categories and ways to implement it in different settings or with different types of patients. However, this embraces only a portion of ICF impact: changing the paradigm of how disability was perceived and addressed, is the main achievement of ICF. Such a change moves from the theoretical perspective the ICF embraces, i.e. the biopsychosocial model, to finally get to the way in which the ICF can be implemented to make the paradigm change possible: this means implementing the ICF in administrative and statistical systems. However, and this is a kind of information which is hard to capture through an analysis of the published literature, our survey revealed a lack of capturing this information in administrative and statistical systems. The reasons are likely not technical, as instruments to implement the ICF exists, so this is a further implementation step which need to be targeted by countries and CCs.]